# Caregiver transformation and relational growth in a parent-mediated intervention for autism in Hong Kong – A qualitative study

Camille K. Y. Chan[1], Cecilia H. M. Wong[1], Paul W. C. Wong[2*]

**1** Faculty of Social Sciences, The University of Hong Kong, Hong Kong, **2** Department of Social Work and Social Administration, Faculty of Social Sciences, The University of Hong Kong, Hong Kong

* paulw@hku.hk

## Abstract

Parent-mediated interventions (PMI) have demonstrated effectiveness in supporting autistic children's emotional regulation, social engagement, and communication. However, most research has focused on quantitative child outcomes, with limited attention to caregivers' lived experiences, particularly within non-Western urban contexts such as Hong Kong. This study explored how a culturally adapted PMI shaped caregiving practices and parent-child relationships among Hong Kong caregivers of autistic children. Using a qualitative design, five focus groups were conducted with 22 caregivers of children diagnosed with or suspected to be on the autism spectrum. Reflexive thematic analysis identified two central themes: (1) Caregiver transformation, capturing shifts in caregivers' mindsets, emotional regulation, reflective parenting, and interactional practices as they moved from being supporters to co-regulators; and (2) Cultivating togetherness, encompassing enhanced parent-child synergy through deeper emotional attunement, mutual responsiveness, and relational engagement. Caregivers described a shift from technical caregiving to authentic relational engagement, marked by greater sensitivity to their children's needs and strengthened reciprocal dynamic in daily interaction. Structured guidance and individualised feedback were consistently identified as critical enablers, providing practical, context-sensitive strategies suited for small living environment. Furthermore, children were increasingly recognised as relational stakeholders rather than passive recipients of care. Some caregivers noted ripple effects, with other family members adopting supportive practices that fostered more cohesive family environments. This study provides qualitative evidence that culturally adapted PMI can encourage caregiver transformation, promote well-being, relational attunement, and developmental pathways that affirm neurodivergent identities. Findings underscore the value of inclusive, family-centred approaches that nurture mutual understanding and long-term relational well-being.

**Data availability statement:** The data from this study are available upon request due to ethical restrictions on sharing data publicly. Data can be provided upon reasonable request from the ethics committee at our university - Human Research Ethics Committee, The University of Hong Kong: hrec@hku.hk (quote reference: EA240065).

**Funding:** This study was funded by the Hong Kong Jockey Club Charities Trust (2015-0081-011 to PW). The funder had no role in study design, data collection and analysis, decision to publish, or preparation of the manuscript.

**Competing interests:** The authors have declared that no competing interests exist.

## Introduction

Families of children with Autism Spectrum Disorder (ASD) face multifaceted developmental challenges that affect their emotional well-being, relationship, and daily lives [1]. ASD is defined as a complex neurodevelopmental condition characterised by differences in social communication, interaction, and restricted or repetitive behaviours, all of which can vary in intensity and expression across individuals [1]. Caregivers of autistic children in Hong Kong often experience additional stress due to socio-cultural expectations and environmental constraints [2]. The strong emphasis on academic achievement and social conformity imposes high demands on children, while small living space, high living costs, and constrain access to professional resources intensify parental stress [3,4]. Caregivers may internalise the stigma of their child's neurodivergence, viewing it as a personal or familial shortcoming [3]. This underscores the need for culturally sensitive interventions that promote caregiver well-being alongside child development. In spite of governmental provision for early identification, referral, and specialised education [5], Hong Kong families still experience substantial delays in accessing assessments and service—ranging from 13 to 19.6 months [6], consistent with global average timelines [7,8]. During this waiting period, caregivers are often left to navigate uncertainties with limited support, while managing intergenerational differences in parenting styles and experiences of affiliate stigma [9,10].

To address these needs, a culturally adapted parent-mediated intervention (PMI) was introduced to support Hong Kong caregivers raising autistic children. The intervention evaluated in this study was adapted from the World Health Organization Caregiver Skills Training (WHO-CST) program. This program is designed to improve the confidence, parenting skills, and overall well-being of caregivers raising children with developmental delays by using evidence-based parenting and PMI strategies [11]. It was introduced locally in 2018 through the Jockey Club Autism Support Network and further localized in 2021 by the local NGO master trainers to reflect Hong Kong's specific cultural and caregiving context [12]. The programme teaches caregivers of children aged two to six with responsive caregiving strategies such as developing routines and introducing tools for emotional attunement and stress management. Across nine structured group-based sessions, caregivers are guided to integrate practical approaches into daily play and home activities, which becomes opportunities for child development. Rather than positioning the caregivers as therapists, the training empowers them to be responsive, confident, and emotionally attuned with their child's unique developmental journey. Key objectives of the programme include strengthening caregiver-child interactions, supporting children's learning and communication, enhancing caregiver well-being. The programme explored in this study is grounded in the concept that these elements work together to reshape caregivers' perspectives, thereby enabling them to not only navigate challenges but also to foster their child's identity and potential.

Quantitative studies on PMIs have documented promising outcomes related to parental adherence, child development, and parental skill [13,19,21], yet, fewer studies have explored the programme participation experiences of caregivers that shape how these

changes occurred. Understanding the emotional, cognitive, and relational shifts experienced by caregivers is critical to building more responsive and sustainable intervention models. This paper addresses this gap by exploring the lived experiences of caregivers in Hong Kong who participated in this culturally adapted PMI. Specifically, it investigates how the intervention influenced caregivers' mindsets, parenting practices, and relationships with their children. By focusing on these qualitative changes, the study seeks to uncover the relational and emotional mechanisms that drive caregiver transformation.

## Materials and methods

### Ethics statement

Ethical approval was obtained from the Human Research Ethics Committee of the University of Hong Kong (reference number: EA240065), adhering to standards for human participant research. All participants received detailed information on study objectives, procedures, and their rights before given electronic informed consent, supplemented by verbal reminder by researchers before each focus group discussion. Participant confidentially was maintained through de-identification of data, and participants were informed of their right to withdraw at any time without consequence.

### Study design

This qualitative study, grounded in a hermeneutic constructivism [22] and interpretivist epistemology [23], explored caregivers' lived experiences after participating in a parent-mediated intervention designed to support caregivers of children diagnosed with, or suspected to be on, the autism spectrum. A constructivist ontology acknowledges that while certain phenomena may be objectively observed, human experience is shaped through subjective interpretations, which is inherently influenced by individual perspectives, cultural contexts, and social interactions. The interpretivist epistemology emphasises the importance of meaning-making and the diverse ways in which caregivers understand and respond to their experiences, particularly within Hong Kong's sociocultural context. This approach enables a contextually grounded understanding of how caregivers made sense of the intervention, highlighting relational and emotional dimensions that extend beyond quantitative or behavioural outcomes.

The PMI examined in this study was a culturally adapted program based on the WHO-CST model. Further details of the intervention's structure are published elsewhere [12,24]. Each group cohort comprised approximately six to eight caregivers, and the program consisted of nine structured sessions, each lasting between 90 and 120 minutes [25]. This manuscript follows the Consolidated criteria for Reporting Qualitative research checklist [26] for transparency and rigour in qualitative reporting quality (S1 File).

The research team comprised both male and female members, including clinical psychologist, counsellor, and qualitative researcher who were experienced in working with autistic children and with healthcare providers, ensured rigorous study execution. These researchers had not established relationships with the participants prior to study commencement.

### Participants

Participants were caregivers who had completed the above-described intervention and were recruited through purposive sampling. Recruitment was facilitated by partner NGOs, who shared study information with eligible caregivers. Interested participants were guided to an online platform, where they received the research information sheet, provided electronic consent, and completed a background questionnaire indicating the year they participated the programme and the age of their child. Participation was voluntary, and all eligible consenting caregivers were included in the study. Sample size was determined by participant availability and logistical feasibility.

### Data collection

Data were collected in March 2024 through five semi-structured focus group discussions in Cantonese language, each lasting from 60 to 70 minutes, with no more than eight participants, to foster a collaborative environment for sharing

experiences. The focus groups were conducted either face-to-face at the research team's university or at the partnering NGOs' centre, or via Zoom, based on participant preference and availability, and were audio-recorded and transcribed verbatim for analysis.

The discussion guide (S1 Text), developed by the research team, was informed by research on affiliate stigma, the benefits of structured support, the importance of practical parenting skills and reflective practices [4,12]. It also considered generational differences in parenting styles and lifestyle factors to ensure a comprehensive exploration of caregiver experiences. Open-ended questions focused on caregivers' experiences with the intervention, including changes in parent-child relationships, perceived benefits, and challenges encountered.

## Data analysis

All focus groups discussions were transcribed verbatim in the language of discussions. Data analysis followed Braun and Clarke's reflexive thematic analysis using their six-phase framework [27]. The process began with the first author of this paper repeated listening to the audio recordings and reading of transcripts to immerse the research team in the data. Initial codes were manually and deductively created to capture patterns in caregiver experiences, followed by being grouped into preliminary themes that reflected broader data patterns. Codes and themes were refined to align with study objectives, providing a nuanced understanding of the intervention's impact on parent-child relationships and family dynamics. Last, these themes and sub-themes are defined and reported. The Joint Engagement Rating Inventory framework [28] was used as a guide for naming themes, as it offers a structured understanding of engagement and interactional dynamics between caregivers and children.

NVivo software [29] was used to organise and manage data coding. Researchers deliberately avoided using NVivo's auto-coding feature and instead created all codes manually to maintain direct engagement with the data, allowing for a nuanced interpretation based on the participants' lived experiences.

To ensure reflexivity and minimise researcher bias, the research team were engaged in regular discussions throughout the data analysis process, addressing potential influences on interpretation. This reflexive approach allowed for interpretive flexibility, as the researchers continuously revisited and refined themes to ensure they captured the nuanced impact of the intervention on family dynamics. The first author translated the participants' quotations into English during manuscript preparation, and the translation was approved by all authors.

## Results

A total of 22 participants who completed the caregiver training between 2018 and 2024 joined the focus group discussions and participants comprised of 18 mothers, one father, one aunt, one foster mother, and one grandmother. Fourteen out of 22 participants were homemakers. The composition of each focus group is included in Supplementary Materials (S1 Table). Participants' background information, including age, marital status, living arrangement, education level, monthly household income, number of children, and the age of their autistic child, is presented in S2 Table (S2 Table).

The thematic analysis revealed two main themes related to parent-child relationships, which are [1] *Caregivers' paradigm shift: From care providers to persons with growth* and [2] *Changes of children status: From care recipients to stakeholders*, each with two associated sub-themes as detailed in S3 Table (S3 Table). This paper delves into the intervention's influences on caregivers, the child's development, and the evolving dynamic between them.

### Theme 1: Caregivers' paradigm shift: From care providers to persons with growth

The first theme, *caregivers' paradigm shift: From care providers to persons with growth*, highlights the evolution of caregivers' perspectives and parenting skills during the intervention. Participants described significant shifts in their approach to caregiving, marked by greater empathy, intentionality, and awareness of their child's needs. This paradigm shift reflects both the

changes in mindset and the acquisition of practical strategies, captured through two sub-themes: *from knowledge-based to practice based* and *from technical to authentic engagement*.

### Sub-theme 1.1: From knowledge-based to practice-based

*This sub-theme* reflects caregivers' evolving understanding of their child's needs, characterised by greater awareness of their roles. It represents caregivers' transformative journey, highlighting their accounts of learning from the intervention, integrating this knowledge, and applying it in their interactions with their child. This process also involves self-reflection, which promotes personal growth and cultivates a positive attitude towards the child.

Almost all participants described that they had adopted a more child-centred perspective, characterised by greater awareness of their child's emotions and individuality. Caregivers in all five focus groups noted a transition in their approach, moving from a parent-centred parenting model to one that recognises and accommodates the unique needs of each child. For instance, many reported observing their child more closely, allowing for freer play, and creating environments that encouraged independent exploration. A mother shared, *"What struck me the most was the first step, 'paying attention' (to the child's behaviours), which is something I hadn't really considered before. You might not have noticed their (the child's) preferences or the things they like to do. When you cater to their interests, you can optimise their learning"*. Another participant mentioned, *"Before the classes, I wouldn't specifically set up a small table and some chairs at home (for play)... After the classes (I set up this space as suggested by the facilitator)… we sit together in the chairs and focus on playing with toys. This change has made us both happier when spending time together"*. These insights demonstrate the ongoing learning process where caregivers integrated new knowledge from the intervention into their daily interaction with their child.

Caregivers also reported self-reflection and personal growth as positive outcomes of the intervention. Many reported heightened self-awareness, recognising that their actions and emotional states could directly influence their child's behaviours and feelings. One participant noted, *"you start to notice certain things may trigger his (the child's) anger and realise there may be underlying reasons for it. You were just playing with him, but you didn't expect that this play could cause him frustration because you may have interfered with his play. And then you wonder why he's upset or misbehaving"*. While the term 'misbehaving' may not align with a neurodiversity-affirming perspective, its use by the caregiver reflects both a developing awareness of the child's underlying needs and emotional triggers, while also reflecting a common parental language in Hong Kong for unexpected behaviour. Participants frequently emphasized the importance of self-care and the value of appreciating their own effects as parents and caregivers. One explained that being a parent of a child with special educational needs can be challenging and noted, *"One of the great aspects of this intervention is it emphasises on the importance of taking care our own emotions as parents. When you are happy, your child will be happy, too. Our emotions can directly impact our child"*.

The intervention also fostered improvement in caregivers' attitudes towards their children. Across all focus groups, participants described feeling calmer, more patience, and more intentional in their interactions with their child. Many reported dedicating time to their child after the program have led to stronger connections and a more fulfilling relationship. These shifts in attitude appeared to have laid the groundwork for a more supportive and nurturing relationship with their children, as well as fostering a sense of accomplishment for being able to provide appropriate care their child needs.

### Sub-theme 1.2: From technical to authentic engagement

This sub-theme emphasises how caregivers authentically engaged with their child, leveraging their newly acquired technical skills. It delves into four aspects, including developing communication and language skills, structured guidance and behavioural management, emotionally regulated caregivers, and commitment and consistency in practice.

Participants reported improvements in engagement with their child through verbal communication, emphasising the importance of proactive and open dialogue. Many described that establishing clear rules early on and informing these rules while encouraging children to communicate their wants and preferences are useful strategies. A father shared that,

*"we pre-informed him about the rules. He (Our child's) is now less reluctant to holding our hands when we go out. He used to avoid touching hands and would run off as soon as he left the house. While it may not always work, we noticed significant improvement as we get to hold hands for a longer period each time."* A mother shared the facilitator's advice on expanding her child's vocabulary, *"the facilitator taught us to engage our children by discussing their interests. For instance, if he is playing with a red car, you can say, 'Tom is playing with a red car,' and then observe his reaction. If he doesn't respond, you might add, 'What kind of car is this? Oh, it's a red taxi!'"*, which they can add into their play routine. Furthermore, another participant stated that, *"in the past, I used to grab and pass on items he pointed at... Now I make sure he verbally asks for what he wants… so that he articulates his needs"*.

Caregivers also reported child's behavioural improvements under structured guidance. They learned to provide appropriate disciplinary actions and setting realistic expectations for their children. A mother recounted how her child used to throw toys during moments of distress and described applying the facilitator's advice to introduce gentle boundaries to help the child process these situations. She shared, *"I told my child, 'This toy is going to jail because you threw it'. I placed the toy on a high shelf where she could see it, and she kept looking at it all day... After that, she no longer threw toys when she got upset... She learned that misbehaviour has consequences"*. Furthermore, participants found it valuable to learn how to start with achievable tasks and to subsequently break down larger tasks into smaller steps for their children. A mother explained, *"we should start from the child's perspective... As the intervention suggested, breaking each task into smaller steps allows us to practice observing and imitating what our child do"*, which she found essential in supporting her child's learning process and connecting with them more effectively.

All participants recognised the benefits of regulating their own emotions, particularly highlighted mindfulness and meditation techniques in improving their calmness during challenging moments. One participant learnt that, *"when we get frustrated by our children, we can practice mindfulness exercises, but it doesn't necessarily have to be mindfulness. The key is to find a way that works best for you to relax"*. Many participants recognised their emotional expression can also serve as a role model for their children. One explained that, *"the facilitator taught us the Emotional Traffic Light technique, emphasising that we should pay attention to the child's facial expressions and behaviours to understand their emotional states. We want to avoid adding fuel when they are at yellow because it can become uncontrollable when it reaches 'red'… I even made a printout of a traffic light to teach my child at home, saying, 'if you see mum looking like this, please let me know'"*. Through these practices, caregivers learned to understand their own and their child's emotions, fostering co-regulation and modelling calmer responses during emotionally charged situations.

Participants also emphasised the importance for consistent practice and long-term commitment in applying the skills they had learned. Nearly all participants recognised that the continuation of these practices demanded ongoing effort, and they expressed confidence in their ability to incorporate the techniques into their daily routine, which requires a genuine commitment to engage with their children. However, they also expressed the additional benefits if follow up or refresher courses were available. One participant stated that, *"completing this course doesn't mean we have grasped everything. I believe the challenge lies in integrating all the techniques as we are still in the process of learning many of them... it comes down to whether parents remember using them in their daily lives and incorporate them into everyday scenarios"*.

Overall, the theme *"Caregivers' paradigm shift: From care providers to persons with growth"* shows that acquiring practical skills can lead to meaningful transformation in caregivers' parenting perspectives. The narratives of the caregivers demonstrate the potential of self-reflection for a broader process of growth in caregivers' roles, building their confidence and resilience in supporting their child's development, thereby strengthening the parent-child relationships.

## Theme 2: Changes of children status: From care recipients to stakeholders

The second theme, *Changes of children status: From care recipients to stakeholders*, reflects a transformation wherein caregivers view their child not as passive care recipients, but as active participants in their relationships. The caregivers explained the potential paradigm shift within themselves in understanding their child's identity, as they observed increased

responsiveness from their child, particularly in emotional expression, independence, and trust. This shift corresponded with caregivers' perception of a more reciprocal and connected relationship as reflected in positive changes in both behaviour and parent-child interactions. The theme encompasses two sub-themes: *Fostering fluid and connected relationships* and *Parent-child togetherness*.

### Sub-theme 2.1: Fostering fluid and connected relationships

This sub-theme reflects positive changes participants observed in their child's emotional, social, and behavioural development as they began viewing their child as a stakeholder in their relationship. Participants observed this shift in the child's enhanced emotional expression and regulation, increased joint engagement, and greater compliance.

Participants observed improvement in their child's emotional expression and self-regulation, which is interpreted as a hallmark of a balanced and reciprocated parent-child relationship. Many caregivers reported feeling empowered to validate their child's emotional experiences, strengthening mutual understanding. One mother shared, *"he (my child) used to say, 'I'm scared, I don't want to go to school'. So, I acknowledged his feelings by saying, 'I know you're nervous and scared. You may be afraid of not being able to see me, but I'll be here waiting for you'. Once he learnt that I understood him, he stopped saying it. Although he couldn't articulate his fear, he became more stable in my validation of his feelings"*. This ability to recognise and respond to the child's emotions marked a fluid dynamic, where children's communication prompted supportive responses from caregivers.

Caregivers reported observing reduced dependency and increased willingness to communicate and interact with one another, suggesting the intervention promoted independence and social engagement. For instance, a mother shared that, *"he (my child) preferred to play alone, so much that he didn't want me to interrupt him. But now, he invites me to play with him. He used to draw on his own, but now he invites me to sit next to him and asks for my opinion, such as, 'Which colour should I pick?'"*. Some participants described how their children exhibited greater engagement with other family members, demonstrating cues to greater confidence and sociability. Another participant shared, *"his (my child) play skills and interactions have improved… When someone says 'hello' to him, he would respond with a 'hello'… It's as if he is no longer in his own world, no longer plays in isolation… he now engages more with others, and his interactions have improved"*.

### Sub-theme 2.2: Parent-child togetherness

*Parent-child togetherness* highlights the engagement and interaction quality between the caregiver and child, fostering a sense of connection and mutual enjoyment. This sub-theme emphasises the importance of emotional communication, trust, and shared understanding, as participants' experiences found these elements encourage active participation in daily routines and joint activities. Through strengthened bonds, caregivers experience deeper relational engagement with their child.

Participants largely agreed that open communication between parents and children is important for building trust and creating a safe environment where children feel comfortable sharing their emotions. One participant reflected, *"when she (my child) sees me being angry, she would now says, 'Mom, you're angry.' And I would respond, 'Aren't you also angry?' We both understand what it means to be angry and would agree to step away for five-minutes before resuming the task on hand. This is much better than us arguing"*. The participant further noted that this practice of emotional openness has strengthened their bonds.

Participants also described how shared activities helped fostering meaningful engagement and trust with their child. Structured yet enjoyable activities, like learning through play or creating routines together, were seen as opportunities to deepen their bonds. One participant suggested, *"It's not just about communicating and building relationships through playing with toys, but also about new knowledge we experience in daily life… due to the pandemic and the lockdown in the past few years… their (children's) understanding of the world is largely shaped by what they see on television or in their toys, and they have little concept of the physical world… now that we can go out, we can truly engage with the actual objects,*

*which in turn helps help understand things better through descriptions and conversations"*. A few participants articulated that these shared experiences helped strengthen their parent-child relationship while promoting collaboration and trust.

Overall, this theme illustrated the hypothesized cascading effects of caregiver transformation on children. In particular, caregivers responsiveness and attunement were perceived to associate with changes in their child's emotional expression, independence, and social engagement. This perceived reciprocal growth highlights a potential pathway through which their newly adopted parenting approach may have positively influenced both individual and relational dynamics, fostering a deeper and more collaborative parent-child relationship.

## Discussion

The study explored the lived experiences of caregivers of autistic children who participated in a culturally adapted parent-mediated intervention tailored in Hong Kong. Our findings highlight a dual transformation as described through participants' narratives. First, caregivers reported a shift from task-focused roles to more reflective, growth-oriented approaches, capturing their enhanced emotional regulation, self-care, and interactional skills. Second, caregivers perceived children as increasingly engaging as relational stakeholders within their families, with descriptions of children showing stronger emotional regulation, social engagement, and reciprocal communication. Together, these themes reflect a deepening synergy in the parent-child relationship as perceived by participants and shaped by the intervention.

Participants commonly reported a transformation in their parenting approaches, marked by a shift toward more empathetic, responsive, and child-centred practices. Of the 22 participating caregivers, 10 had their child being diagnosed before the age of two, with 14 identified as homemakers. This underscores the importance of early support at critical developmental windows [30–32], particularly for caregivers who are primary figures in a child's developmental journey. Nearly all participants reported their children became more confident in expressing needs, preferences, and emotions. This aligns with our earlier findings during the adaptation and pretesting phase of the program, which reported improvements in caregiving skills, emotional self-management, handling of challenging child behaviours, and the perceived value of home visits [12]. Our findings are also consistent with findings from international studies, showing that parent-mediated interventions can enhance child engagement, expressive and receptive communication both verbally and nonverbally, and social interactions [19].

Beyond acquiring the taught skills in the programme, caregivers reported personal growth through mindfulness, self-care practices, and exercises in emotional regulation. Within Hong Kong's traditional cultural contexts, caregiving is often shaped as a process of monitoring children to excellent academic achievement, social conformity, and idealised behavioural standards, commonly encapsulated by the phrase "nurturing children to become dragons" [4], which can lead to internalised stigma and feelings of inadequacy when their child's development diverges from expected [4,10]. While the clinical classification of ASD provides diagnostic clarity, the neurodiversity framework highlights autism as a natural variation in human development, valuing differences over deficits, and promoting acceptance and respect [2]. However, some caregivers have yet to adopt neurodiversity-affirming language, possibly influenced by cultural norms and limited familiarity with this perspective.

By joining this adapted programme, caregivers began to recognise and attend to their own emotional needs, shifting their perspective of caregiving from solely child-centric to a fluid and relational process of mutual growth. This shift contributed to more connected and flexible parent-child relationships a more consistent practices of positive parenting. These findings align with previous research highlighting the importance of self-reflection, emotional attunement, and compassionate caregiving [33–35]. Furthermore, many caregivers described their children not as passive care recipients, but as active participants in family life. Our research indicated that parenting programs broadly enhance children's social, emotional, and behavioural development across clinical and non-clinical populations [36], suggesting that such interventions may benefit families beyond the neurodivergent community by equipping caregivers with developmentally supportive strategies.

This relational shift echoes broader trends in naturalistic intervention approaches, which are increasingly recognised for their ability to foster meaningful parent-child interactions while promoting resilience and mutual respect [13–16]. These interventions emphasise dyadic strategies such as synchrony, responsiveness, and joint attention, often reinforced through parent coaching and video feedback [17]. While behavioural improvements may support children in navigating societal norms [18–20], the primary goal of these interventions is not behavioural conformity but relational connection. For the caregivers in our study, the focus on connection rather than correction aligned with a growing recognition of their children's emotional needs and communicative intentions. In this way, the program not only helped address the stressors of caregiving but also embodied neurodiversity-affirmative values that validate children's authentic ways of being.

Several participants also reported changes beyond the dyadic relationships and extend to the wider family dynamics, noting that spouses and grandparents began adopting supportive techniques introduced in the program. These broader family-level shifts are documented in a complementary study by our team, which extends the present findings by examining how non-participating family members responded to the intervention at home [37].Studies outside the autism context have shown that family involvement in caregiver-mediated interventions can improve outcomes for both caregivers and care recipients [38]. However, global implementation of family-inclusive strategies remains limited due to systemic barriers such as insufficient planning and poor integration within healthcare systems [39]. These challenges reflect a broader disconnection between the evidence supporting family engagement and its practical application.

The program also served as a critical support buffer for caregivers long waits for assessment and services in the Hong Kong's public healthcare system [40]. While such interventions cannot replace traditional clinical consultation and care, participants described the structured guidance and individualised feedback as instrumental in reducing uncertainty and enhancing their self-efficacy as caregivers [41,42]. Home visits further reinforced the intervention's relevance by acknowledging spatial constraints and providing context-specific support. For future implementation, it will be important to address systemic and cultural barriers, particularly for ethnic minority families who may face intersecting challenges such as language differences and cultural stigma [43,44]. Culturally competent facilitators and multilingual materials can improve inclusivity and trust. Although prior quantitative findings show limited outcome variation across caregiver backgrounds, this study reinforces the importance of culturally responsive approaches in sustaining engagement [45].

While the findings are promising, the study has limitations. The voluntary nature of caregivers' participation may have led to the inclusion of caregivers who had more positive experiences with the program, as well as not having evaluated changes from the child's perspective. Second, depending on pandemic-related restrictions and service availability, some groups received the intervention through in-person classes with home visits, while others participated via Zoom and submitted video-based exercises for facilitator feedback [24]. This may have influenced how caregivers experienced support and applied strategies at home. Additionally, the reliance on self-reported data may introduce potential bias. Though qualitative research is not intended to produce generalisable results, the depth of participated caregivers' narratives captured meaningful insights into the relational mechanisms that shaped caregiver transformation. This study adds to growing evidence that centres caregivers as active partners in intervention processes and highlights the importance of family engagement in supporting neurodivergent children. It also underscores the need to extend research beyond parent-child dyads to capture the full spectrum of relational and systemic changes sparked by parent-mediated interventions. Future research should explore how these transformation evolve over time through longitudinal and mixed-methods designs.

## Conclusion

This study explored the experiences of caregivers of autistic children in Hong Kong who participated in a culturally adapted parent-mediated intervention. This study also sheds light on underlying philosophical and cultural tensions in the design of autism interventions. While the intervention drew from behavioural principles to guide caregiver-child interactions, its embraced person-centred and neurodiversity-affirming approaches that emphasised acceptance, understanding, and relationship-building [24,50]. This dual approach, both drawing on and resisting normative expectations, resonates

with the internal conflict experienced by caregivers negotiating between societal norms and their child's individual needs. It also reflects a broader tension between the behavioural paradigms often used in clinical models, and the interpretivist and relational orientation of qualitative research [46]. These tensions not only shape the experiences of neurodivergent individuals and their families, but also challenge researchers and practitioners to adopt inclusive, reflective practices within complex sociocultural settings [47]. Recognising and engaging with these tensions may foster more culturally attuned flexible, and family-centred approaches even as these debates continue to evolve.

Building on these findings, we suggest that this culturally adapted intervention may serve as a reference for future parent-mediated programs. Research has consistently shown that addressing both parenting skills and caregiver well-being are more likely to support sustainable outcomes [16,48]. Our findings suggest future iterations should incorporate family-wide strategies, such as co-learning workshops and inclusive resources tailored to different household members. Importantly, caregivers in this study did not describe improvements in their children as becoming "more typical," but rather as becoming more expressive, confident, and engaged. This aligns with a scoping review identifying relational connections and social support as critical to autistic well-being [49].

## Supporting information

**S1 File. COREQ checklist.**
(PDF)

**S1 Table. Focus group characteristics.**
(DOCX)

**S2 Table. Participants characteristics.**
(DOCX)

**S3 Table. Themes and sub-themes.**
(DOCX)

**S1 Text. Discussion guide.**
(DOCX)

## Acknowledgments

We extend our gratitude to the Hong Kong Jockey Club Charities Trust (HKJC) for their support, which made the necessary training for the development, delivery, and programme evaluation possible. Their initiative, JC A-Connect, managed by the Faculty of Social Sciences of the University of Hong Kong, provides essential holistic support for ASD children and their families.

## Author contributions

**Conceptualization:** Camille K.Y. Chan, Cecilia H.M. Wong.

**Data curation:** Camille K.Y. Chan, Cecilia H.M. Wong.

**Formal analysis:** Camille K.Y. Chan, Cecilia H.M. Wong.

**Funding acquisition:** Paul W.C. Wong.

**Investigation:** Camille K.Y. Chan, Cecilia H.M. Wong, Paul W.C. Wong.

**Methodology:** Camille K.Y. Chan, Cecilia H.M. Wong, Paul W.C. Wong.

**Project administration:** Camille K.Y. Chan, Cecilia H.M. Wong.

**Resources:** Cecilia H.M. Wong.

**Software:** Camille K.Y. Chan.

**Supervision:** Cecilia H.M. Wong, Paul W.C. Wong.

**Validation:** Camille K.Y. Chan, Cecilia H.M. Wong.

**Writing – original draft:** Camille K.Y. Chan.

**Writing – review & editing:** Cecilia H.M. Wong, Paul W.C. Wong.

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
