## [Decision Letter · Decision Letter 0]

19 Mar 2025

PMEN-D-25-00020

The impact of parent-mediated interventions on the parent-child relationship in caregivers of children with autism in Hong Kong – A qualitative study

PLOS Mental Health

Dear Dr. Wong,

Thank you for submitting your manuscript to PLOS Mental Health. After careful consideration, we feel that it has merit but does not fully meet PLOS Mental Health’s publication criteria as it currently stands. Therefore, we invite you to submit a revised version of the manuscript that addresses the points raised during the review process.

In addition to the comments by reviewers, I would ask you to please focus on these points during your revision: 

Which language were the discussions with parents conducted in? Did any part of the text analysis use natural language processing? Why or why not?Please include more details regarding the parent-mediated interventions and the caregiver training. Additional details could be provided regarding "hermeneutic constructivism" and "interpretivist epistemology". The former is described in the current manuscript as acknowledging that "reality is socially constructed". Some attention should be given to this description, which seems to endorse a view that objective truth does not exist -- is this appropriate for a scientific journal? Please verify whether the deidentified text transcripts can be included to satisfy the PLOS Data Policy. If data sharing is permitted, then public archiving is preferred over "reasonable request" gatekeeping. Where feet are used as units, please also provide an equivalent measurements in meters.  

We look forward to receiving your revised manuscript.

Kind regards,

Joel Frohlich, PhD

Academic Editor

PLOS Mental Health

Reviewers' comments:

Reviewer's Responses to Questions

**Comments to the Author**

1. Does this manuscript meet PLOS Mental Health’s publication criteria?

Reviewer #1: Partly

Reviewer #2: Yes

2. Has the statistical analysis been performed appropriately and rigorously?

Reviewer #1: I don't know

Reviewer #2: N/A

3. Have the authors made all data underlying the findings in their manuscript fully available (please refer to the Data Availability Statement at the start of the manuscript PDF file)?

Reviewer #1: No

Reviewer #2: Yes

4. Is the manuscript presented in an intelligible fashion and written in standard English?

Reviewer #1: Yes

Reviewer #2: Yes

Reviewer #1: The authors' approach to including a thematic analysis is valuable given the need for more qualitative work/lived experience in the context of psychopathology. I am not an expert in qualitative analyses so I would defer to the expertise of other reviewers in terms of the methodological rigor of the approach taken. However, the limited novelty in the specific questions leveraged coupled with other concerns (small sample size, limited generalizability to different family members, a lack of a counterfactual to compare differences between those with and without intervention, transferability of the current findings to other contexts, etc.) made me waver on the added value of publishing this work over others.

Reviewer #2: The article presents a well-structured and well-written qualitative study into the experiences of caregivers of Autistic children. Changes in caregivers’ perception of their relationship with the child after a parent-medicated intervention are discussed within the cultural context.

The study documents first-person accounts of what works for promoting a positive and inclusive environment within the family of Autistic children, surrounded by a less inclusive society that values achievements and compliance with social norms. These accounts deserve to be heard and published, and the paper contributes to this goal in a rigorous scientific manner.

To improve the paper, I suggest enhancing the discussion that currently remains very close to the study findings and would benefit from a broader reach into the research and social context. This can be done by comparing the findings with other studies of parent-mediated or relation-focused interventions, including from different cultures. Another way of enriching the discussion is by discussing the study within the neurodiversity-affirmative movement, relevant to the study focus.

The paper itself showcases the difficulties of navigating the social context while valuing an Autistic child’s needs and agency. It expresses a neurodiversity-affirmative mindset using a deficit-based language (‘with ASD’) that is compliant with the professional social norms set by the DSM. It is focused on a family intervention empowering the child but views the goal of this intervention as that ‘improved ASD child behaviour, facilitated by parent-mediated interventions, supports smoother integration into societal norms, ultimately promoting social inclusion.’ This perspective links inclusion to the child's compliance with societal expectations, while in Western cultures, inclusion is viewed as the societal environment accommodating diverse ways of being.

Not surprisingly, the text, challenging cultural norms while complying with them, reflects the cultural conflict addressed by the studied intervention, as well as the conflict between the behavioural philosophy of medicine and the person-centred philosophy of qualitative research. These controversies, affecting not only Autistic individuals and their families but also professionals supporting and researching them, deserve to be addressed in an open and reflective manner.

**Do you want your identity to be public for this peer review?** For information about this choice, including consent withdrawal, please see our Privacy Policy

Reviewer #1: No

Reviewer #2: **Yes: ** Olga Dobrushina

---

## [Decision Letter · Decision Letter 1]

16 Jun 2025

PMEN-D-25-00020R1

Caregiver transformation and relational growth in a parent-mediated intervention for autism in Hong Kong – A qualitative study

PLOS Mental Health

Dear Dr. Wong,

Thank you for submitting your manuscript to PLOS Mental Health. After careful consideration, we feel that it has merit but does not fully meet PLOS Mental Health’s publication criteria as it currently stands. Therefore, we invite you to submit a revised version of the manuscript that addresses the points raised during the review process.

A third reviewer was invited to referee the paper. This reviewer's comments that must be addressed before the manuscript can be considered suitable for publication. 

We look forward to receiving your revised manuscript.

Kind regards,

Joel Frohlich, PhD

Academic Editor

PLOS Mental Health

Journal Requirements:

1. In the online submission form, you indicated that [The discussion guide is available in the Supporting Information file. The availability of additional data is available from the corresponding author upon reasonable request due to ethical considerations related to participant and funder consent for data release.].

a. In a public repository,

b. Within the manuscript itself, or

c. Uploaded as supplementary information.

Additional Editor Comments (if provided):

Reviewers' comments:

Reviewer's Responses to Questions

**Comments to the Author**

Reviewer #1: (No Response)

Reviewer #2: All comments have been addressed

Reviewer #3: (No Response)

publication criteria?

Reviewer #1: (No Response)

Reviewer #2: Yes

Reviewer #3: Partly

3. Has the statistical analysis been performed appropriately and rigorously?

Reviewer #1: (No Response)

Reviewer #2: N/A

Reviewer #3: N/A

4. Have the authors made all data underlying the findings in their manuscript fully available (please refer to the Data Availability Statement at the start of the manuscript PDF file)?

Reviewer #1: (No Response)

Reviewer #2: No

Reviewer #3: No

5. Is the manuscript presented in an intelligible fashion and written in standard English?

Reviewer #1: (No Response)

Reviewer #2: Yes

Reviewer #3: Yes

Reviewer #1: (No Response)

Reviewer #2: (No Response)

Reviewer #3: This paper addresses an important topic – caregiver experiences of a particular parent-mediated therapy – within a particular cultural context. Parent-mediated therapies have an emerging body of evidence in the autism field, albeit most research has focused on child-related quantitative outcomes (as acknowledged by the authors).

There are some aspects to the paper that I feel would need addressing/resolving before this paper would be suitable for publication.

One aspect that I think needs more consideration from the authors is addressing the tension in the paper of referring to working within ‘neurodiversity-affirming approaches and frameworks’ while at the same time, some of the language used in the paper (e.g., “ASD children” – more preferred language amongst the autistic and autism communities is ‘autistic children’) and some of the approaches and outcomes reported that focus on child-related behaviours that were targeted (e.g., lines 263-269 of the clean version, starting at “A mother shared how her child would throw toys during tantrums….”) would not align with this approach. I note that there may be cultural differences in what is considered ‘neurodiversity-affirming’ and that I may be bringing a Western conceptualisation, however, I think it would be important for the authors to address this point if that is the case.

Without more detailed information about the intervention (e.g., size of the groups, length of sessions, who attends the sessions, delivery mode, key strategies, theory of change, active ingredients etc) it is also difficult to ascertain if the entire therapy program itself would align with a neurodiversity affirming framework.

The other aspect to the paper that requires more consideration is around the interpretations and conclusions from the data. There are multiple claims in the paper that I feel are not directly supported by the data being reported on and these need to be more accurately represented for publication purposes. For example:

• After a caregiver quote, the authors conclude “These practices helped caregivers foster a more supportive and emotionally stable environment at home.” – are there any data (beyond caregiver quotes) to support this extension?

• “This shift marks the evolution of their relationship from one-directional to reciprocal” – again, are there any data to support this? I think it could also be asserted that the relationship was always reciprocal, however, caregivers are now reporting feeling more attuned/connected (if I am interpreting the caregiver quotes accurately)

• “Overall, this theme illustrated the cascading effects of caregiver transformation on children. As caregivers became more responsive and attuned to their child’s needs, children demonstrated greater emotional expression, independence, and social engagement. This reciprocal growth highlights how the intervention positively influenced both individual and relational dynamics, ultimately fostering a deeper, more collaborative parent-child relationship.” Without a mediation/mechanism analysis to support this claim, this should only be presented as a hypothesised pathway. I do not think this statement is supported by the available data.

• “Our findings revealed a dual transformation: caregivers experienced a paradigm shift from task-focused care providers to reflective individuals engaging in personal growth, while children’s roles evolved from passive recipients to relational stakeholders within their families.” – without consulting children about their experiences (and without other child-outcome data), I feel this is an interpretation that is not supported by the direct data.

I acknowledge that by tempering these conclusions, it does further limit some of the outcomes/conclusions of the paper.

Finally, while I more than understood the paper, it would benefit from a detailed grammatical review by the authors to address the grammatical errors throughout.

**Do you want your identity to be public for this peer review?** For information about this choice, including consent withdrawal, please see our Privacy Policy

Reviewer #1: No

Reviewer #2: **Yes: ** Olga Dobrushina

Reviewer #3: No

---

## [Decision Letter · Decision Letter 2]

21 Aug 2025

PMEN-D-25-00020R2

Caregiver transformation and relational growth in a parent-mediated intervention for autism in Hong Kong – A qualitative study

PLOS Mental Health

Dear Dr. Wong,

Thank you for submitting your manuscript to PLOS Mental Health. After careful consideration, we feel that it has merit but does not fully meet PLOS Mental Health’s publication criteria as it currently stands. Therefore, we invite you to submit a revised version of the manuscript that addresses the points raised during the review process.

While the main concerns expressed by reviewers have been addressed, the manuscript is in need of careful editing for grammatical errors before it can be accepted. 

We look forward to receiving your revised manuscript.

Kind regards,

Joel Frohlich, PhD

Academic Editor

PLOS Mental Health

Journal Requirements:

1. In the online submission form, you indicated that [The discussion guide is available in the Supporting Information file. The availability of additional data is available from the corresponding author upon reasonable request due to ethical considerations related to participant and funder consent for data release.].

a. In a public repository,

b. Within the manuscript itself, or

c. Uploaded as supplementary information.

Additional Editor Comments (if provided):

Reviewers' comments:

Reviewer's Responses to Questions

**Comments to the Author**

Reviewer #2: All comments have been addressed

Reviewer #3: All comments have been addressed

publication criteria?

Reviewer #2: Yes

Reviewer #3: Yes

3. Has the statistical analysis been performed appropriately and rigorously?

Reviewer #2: N/A

Reviewer #3: N/A

4. Have the authors made all data underlying the findings in their manuscript fully available (please refer to the Data Availability Statement at the start of the manuscript PDF file)?

Reviewer #2: No

Reviewer #3: No

5. Is the manuscript presented in an intelligible fashion and written in standard English?

Reviewer #2: Yes

Reviewer #3: Yes

Reviewer #2: My comments have been addressed by the previous revision.

Reviewer #3: Thank you to the authors for their considered response to my previous concerns. I feel my main concerns have been addressed through this revised version of the manuscript. I noted only minor grammatical errors that need addressing, of which, I have highlighted below.

Line 33-34: “…context-sensitive strategies for environment with spatial limitations.” – sentence is unclear

Line 39: “…interventions can foster transformative in caregiver change”

Line 69-70: “The programme teaches caregivers of children aged two to six with responsive caregiving strategies…”

Lines 77-80: “These elements work together to shift caregivers’ perspectives—not only helping them manage challenges but also encouraging them to embrace their child’s identity and potential, forms the foundation of the programme explored in this study.” – sentence is unclear

Lines 389-390: “..particularly for caregivers who are primary figures a child’s developmental journey.” Sentence missing the word “in”

Line 395: “…with findings from internation studies” – should this be ‘international”?

Line 416: “..Oruir research indicated…” – should be “Our”

418: “beyong” – should be ‘beyond’?

494: “Consistent with neurodiversity-affirmative frameworks, these approaches shift the focus from behavioural normalisation towards behavioural normalisation toward mutual respect” – duplication?

**Do you want your identity to be public for this peer review?** For information about this choice, including consent withdrawal, please see our Privacy Policy

Reviewer #2: **Yes: ** Olga Dobrushina

Reviewer #3: No

---

## [Editor Report · Decision Letter 3]

29 Aug 2025

Caregiver transformation and relational growth in a parent-mediated intervention for autism in Hong Kong – A qualitative study

PMEN-D-25-00020R3

Dear Prof Wong,

We are pleased to inform you that your manuscript 'Caregiver transformation and relational growth in a parent-mediated intervention for autism in Hong Kong – A qualitative study' has been provisionally accepted for publication in PLOS Mental Health.

Best regards,

Joel Frohlich, PhD

Academic Editor

PLOS Mental Health

I am pleased to see that, as requested by the last round of review, the manuscript has been edited for clarify and grammar. I believe the writing is sufficiently clear to proceed with formal acceptance and publication. Optionally, however, you may wish to check your manuscript for minor grammatical issues besides those pointed out by a reviewer. With respect to clarity and ease of understanding the text, for instance, you may wish to revise line 280 (page 15), which is probably intended to state either "that the content of their emotional expression" or simply "that their emotional expression". Although other smaller grammatical issues are present elsewhere, I leave it to your discression whether or not to have the text checked for grammar errors before proceeding to manuscript production.